# The Amine Functionalized Sugarcane Bagasse Biocomposites as Magnetically Adsorbent for Contaminants Removal in Aqueous Solution

**DOI:** 10.3390/molecules26195867

**Published:** 2021-09-28

**Authors:** Chairul Irawan, Meilana Dharma Putra, Hesti Wijayanti, Rinna Juwita, Yenny Meliana, Iryanti Fatyasari Nata

**Affiliations:** 1Department of Chemical Engineering, Lambung Mangkurat University, Banjarbaru 70714, Indonesia; cirawan@ulm.ac.id (C.I.); mdputra@ulm.ac.id (M.D.P.); hesti.wijayanti@ulm.ac.id (H.W.); rinna.juwita@ulm.ac.id (R.J.); 2Indonesian Institute of Science, Research Centre Chemistry, Tangerang 15314, Indonesia; yenn001@lipi.go.id; 3Wetland-based Materials Research Centre, Research Institute and Community Service, Lambung Mangkurat University, Banjarbaru 70714, Indonesia

**Keywords:** biocomposites, magnetic, Pb(II) ion, solvothermal, sugarcane bagasse

## Abstract

The method of solvothermal by one-step operation has been performed to synthesize of magnetic amine-functionalized sugarcane bagasse biocomposites (SB-MH). The obtained SB-MH contains 62.34% of Fe, 17.8 mmol/g of amine, and a magnetic property of 19.46 emu/g. The biocomposite surface area increased significantly from 1.617 to 25.789 m^2^/g after amine functionalization. The optimum condition of SB-MH used for Pb(II) ion removal was achieved at pH 5 for 360 min with adsorption capacity of 203.522 mg/g. The pseudo 2nd order was well-fitted to the model of Pb(II) ion adsorption. Meanwhile, other contaminant parameters number of Chemical Oxygen Demand (COD), Total Suspended Solid (TSS), and dye in wastewater were also remarkably reduced by about 74.4%, 88.0%, and 96.7%, respectively. The reusability of SB-MH with 4th repetitions showed only a slight decrease in performance of 5%. Therefore, the proposed magnetic amine-functionalized sugarcane bagasse biocomposites lead to a very potential adsorbent implemented in high scale due to high surface area, easy separation, stable materials and capability to adsorb contaminants from aqueous solution.

## 1. Introduction

Recently, the utilization of bio-source materials recovered from biomass has been increasing for research interest. The proper treatment of lignocellulosic materials could produce value-added products such as building block chemicals, sugar, specific polymer, fuel and adsorbent [1,2]. The biomass consisting cellulose, hemicelluloses, and lignin has the capability to adsorb heavy metal ions; and even it can be used as a harmless adsorbent. Furthermore, pre-treatment of biomass with acid solution (nitric acid, sulphuric acid, hydrochloric acid, citric acid) and/or base solution (calcium hydroxide, sodium carbonate, sodium hydroxide) mineral effaces soluble biotic compounds and even may intensify the metal adsorption efficiency [3]. A low-cost farming biomass is sugarcane bagasse. It is a solid part excessed from sugarcane industry. Researchers have applied the cellulose bagasse for biocomposite matrices such as cardanol–formaldehyde composites [4], polyester matrix [5], composite bioplastic [6], and pesticides [7]. The modified sugarcane baggase is also used as adsorbent for dyes removal [8], CO_2_ adsorption [9], and metal ion adsorption [10]. There are many contaminants in waste water, and one of the dangerous ions is Pb(II); it is known as a poisonous substance in water. The elevated Pb(II) ion concentration in drinking water will damage human health, causing things such as mental retardation, kidney failure, physiological flaw, and anemia [11]. Heavy metal removal from industrial and urban discharges can decrease their dangerous impact substantially on biota, environment, animal and human health [12]. Another earnest problem around the world is textile dye; whereas 1–15% of dyes are lost in the emission throughout operation [13]. Dyes are highly poisonous to the watery biota as they are mutagenic and carcinogenic and also cause allergy and skin irritation from a health point of view [14]. Disposal of dyes into water will influence its aesthetic nature and destroy the watery ecosystem due to high concentration of color, Total Suspended Solid (TSS), Biochemical Oxygen Demand (BOD), and Chemical Oxygen Demand (COD) [15].

Several techniques such as ion exchange, reverse osmosis, chemical precipitation, membrane separation, adsorption, and solvent extraction have been used to eliminate heavy metals, dye, and contaminants in waste water [16,17,18,19]. The adsorption process is an effective practice, and a low-cost method for removal of Pb(II) ions from aqueous solutions [20]. However, if the surface modification does not have enough active surface sites, then adsorption is less efficient. Various functional groups as new techniques are developed to make adsorbents, including amine groups, amide, and carboxyl that have elevated selectivity toward poisonous metals [21]. Amine groups are able to absorb a number of metal anions and cations from aqueous solutions due to their specific surface charge [22,23]. 

Our previous studies have concerned the use of amine rice husk magnetic nanoparticles for metal ion adsorption [20,24]. Those studies are in order to increase the adsorption capacity of metal ions and specific surface area created through modifying of sugarcane bagasse cellulose; hence, the materials become biocomposites with robust magnetic property for easy separation. The preparation of amine magnetic sugarcane bagasse biocomposites was designed to improve the capability and efficiency of adsorption and also degrade other contaminants in waste water. In this study, preparation and characterization of amine-functionalized magnetic sugarcane bagasse biocomposites (SB-MH), adsorption toward the Pb(II) ions as well as the kinetic study, the removal of COD, TSS, and dye in wastewater by only a one-step adsorption process were inspected. In addition, the capability of biocomposites for used repeatedly in adsorption was also investigated.

## 2. Results and Discussions

### 2.1. Amine Functionalized Magnetic Sugarcane Bagasse Biocomposites Characterization

Initial material of sugarcane bagasse (SB) was delignified and mixed with solvent and chemicals for biocomposite formation. The grinding process caused the damage of SB surface providing high surface area and material porosity; but lignin was still banded to hemicelluloses and cellulose. The preparation steps of amine magnetic sugarcane bagasse biocomposites is presented in Figure 1. 

The SB structure looks flat and smooth, confirmed by the FE-SEM image as shown in Figure 1a. Furthermore, the breakage of SB from the surface to the inside was inflicted by the process of delignification, as shown in Figure 1b; this is called SB-D. The original structure of sugarcane bagasse had a large amount of extractives; they were removed from the surface after delignification [25]. Consequently, the structure of lignocellulose bonds began to break and further became an irregular structure (circle area, Figure 1b). Figure 1c shows amine magnetic sugarcane baggase biocomposites; the magnetic component appeared on the SB-D surface, further named SB-MH. The SB-MH contained about 62.34% of Fe with apparent associate by magnetic nanoparticles which have a size of around 30–50 nm (Figure 1c, inset). The hexamethylenediamine as amine source for surface functionalization on biocomposites was successfully deposited as detected at about 17.8 mmol/g. The low porosity of SB-D based on the isotherm of the amount of nitrogen uptake was confirmed by a low surface area with a value of only 1.617 m^2^/g. Surprisingly, the surface area of SB-MH significantly increased up to 25.789 m^2^/g; it was about 15-fold higher compared to SB-D. These results were also supported by pore volume analysis distribution; the value of SB-D was much smaller than SB-MH—0.0463 cc/g compared to 0.115 cc/g, respectively. Thus, the apparent magnetism on the surface of SB-D caused a significant increase in surface area. Similar findings with increases in surface area were also obtained through biocomposite surface functionalization as reported [26,27].

To prove that SB-D has high crystal structure, it was determined by XRD. Cellulose crystal could be observed at an angle of 2θ between 20°–40°. Cellulose consists of millions of microfibrils that contain an amorphous section; it is formed from a cellulose bond with flexible masses and crystalline. The part of the crystal is isolated to produce the finest microcrystalline cellulose. The characterization of the peak of SB and SB-D that contained cellulose fiber was identified at 16.78° for amorphous and 21.69° for crystal formation at 2 tetha (°) (Figure 2a). It was also be proven by SB-D that there was a high intensity of the peak compared to SB. Furthermore, due to loss of hemicelluloses and lignin content, the amorphous structure was mostly changed to be crystalline cellulose. The crystallinity index (CrI) of SB-D increased up to 61.54%, and this result was similar to the result of alkaline treatment of sugarcane bagasse with a crystallinity index of about 63.15% [28]. Based on the XRD result, the establishment of magnetic nanoparticles was also identified with appearance peaks at 36°, 43°, and 57° as magnetite of Fe_3_O_4_. Those peaks were matched with the crystalline magnetite according to the standard pattern crystal data (JCPDS card 39-0664).

The FT-IR spectrum was investigated for the functional groups on materials. The peak at 2900 cm^−1^ was detected as the stretching vibrations of C-H for bonding structure of SB-D and SB (Figure 2b). The amine group was identified at 1640 cm^−1^ individually for SB-MH. The Fe-O stretching band for Fe_3_O_4_ was detected at 580 cm^−1^. The wavenumber at 1050 cm^−1^ confirmed the presence of the O-H and Si-OH band and belongs to SB and SB-D. In addition, peaks at 1260 and 1525 cm^−1^ indicated the lignin aromatic for the C=C bond on SB; however, those peaks did not exist on SB-MH. This finding is reasonable because the content of lignin was decreased and cellulose was detached [28].

Furthermore, SB-MH with the magnetite (Fe_3_O_4_) phases has superparamagnetic properties and consequently is able to react with the outward magnetic field. This character will facilitate an easy separation. To prove the saturation magnetization measurement, it is necessary to conduct the SQUID analysis at room temperature. Figure 3a shows the proceeds of ferromagnetic behavior for the naked amine magnetic nanoparticle (MH) and the SB-MH. The magnetic saturation values of MH and SB-MH are 67.94 and 19.46 emu/g, respectively. The reduction number of the saturation value for SB-MH was about 71.4%; this is most likely associated to the cellulose amount in SB. A similar result was also obtained for the rice husk magnetic biocomposites [20,29]. The biocomposite response to the external magnetic field which contains 0.15 g of fiber in 5 mL of water takes 6 s apart from the solution (Figure 3a, inset). The magnetic property of materials may influence the capability as an adsorbent by increasing the adsorption capacity against a large range of biological toxins [30].

The thermal gravimetric method was applied to study the typical degradation of biocomposites carbonized up to 1000 °C. From Figure 3b, the SB-D clearly showed a two-stage pattern with the decrease in weight. The organic compound evaporation occurred in the range of 30–200 °C (1st stage). Then, the weight loss in the 2nd stage was indicated at 200–600 °C as shown for cellulose, lignin and hemicelluloses with the degradation until 95.7%. This typical degradation is similar to that of other lignocellulose materials [31]. On the other hand, the TG curve for SB-MH appears to be quite resistant to temperature, as the evaporation of water only take about 5.15% weight loss at 200 °C for the 1st stage. On further observation, the SB-MH curve was the decomposition of lignocelluloses material and amine groups up to 600 °C with the remaining material of 62.34%. Interestingly, the biocomposites of amine-functionalized magnetic sugarcane bagasse took place in stable conditions after 600 °C; it is plausible due to the presence of magnetic nanoparticles. Even more, the SB-MH remained in a residual amount of weight about 34.86% higher than that for SB-D.

### 2.2. Kinetic Study of Pb(II) Ion Adsorption onto Amine Functionalized Magnetic Sugarcane Bagasse Biocomposites

The kinetic behaviour and adsorption capacity are significant aspects to study the heavy metal ion adsorption process, especially for new potential adsorbents. For this study, the SB-M also was used as a control to observe the effects of amine functionalization on biocomposites to adsorption capacity. The adsorption kinetics models for pseudo 1st (Equation (1)) and pseudo 2nd model (Equation (2)) were applied to study the mechanism of biocomposites [32,33].
(1)$$Q_{t}{=Q}_{e}(1\;-{\;e}^{-k_{1}t})\;$$
(2)$$\frac{t}{Q_{t}}=\frac{t}{Q_{e}}+\frac{t}{k_{2}Q_{e}^{2}}$$

Q_e_ (mg/g) is the adsorbed amount of Pb(II) ions at equilibrium and Q_t_ (mg/g) is the adsorbed amount of Pb(II) ions at time t. The k_1_ (min^−1^) is the pseudo 1st order rate constant and k_2_ (g/mg min) is the pseudo 2nd order rate constant for the adsorption.

The profile of Pb(II) ion adsorption capacity on SB-M and SB-MH is shown in Figure 4. Both types of adsorbents have the ability to bind Pb(II) ions with an increase in adsorption capacity up to 150.628 and 203.522 mg/g for SB-M and SB-MH, respectively. Further, the adsorption capacity became constant, indicated by an equilibrium phase achieved. The rapid increase in adsorption capacity at the beginning process was possible due to the high Pb(II) ion concentration. Furthermore, the availability of empty sites on the adsorbent also caused a high rate of adsorption. The adsorption equilibrium time of Pb(II) ions for both SB-M and SB-MH was achieved in 360 min. At this condition, SB-MH has a larger adsorption capacity of about 25.99% compared to SB-M. The adsorption of SB-M at pH 5 led to deprotonation of the magnetic surface into FeO^−^. The adsorption of Pb(II) ions occurred due to negative charge surface formation and electrostatic interaction to capture the Pb(II) ions [1]. In addition, the surface area of SB-MH was 2.82-fold higher than SB-M. This result proved that the presence amine groups on biocomposites enhanced the adsorption capacity of Pb(II) ions. Due to formation of –NH_3_^+^_,_ the positive surface charge of the magnet was stable. The water solubility of amine increased the hydrogen bonding as well as the lone electron pair, then capturing the Pb(II) ion [24]. This confirms that the amine functionalized on biocomposites has a significant effect on adsorption capacity.

Based on the result of adsorption fitting data (Table 1), the adsorption kinetics of Pb(II) ions for SB-MH is in accordance with the pseudo 2nd order adsorption kinetics model with an R^2^ value of 0.99; it means that the control of adsorption process is chemical adsorption. The mechanism of the pseudo 2nd order adsorption process is described by assuming that the determinant of the reaction rate is a chemical entrapment process including the electron exchange between adsorbate and adsorbent or inter-valence force. This finding is in accordance with the results of other studies [34]. Furthermore, the occurrence of chemical adsorption was mostly related to presence of amine groups on the adsorbent. In addition, the affinity of the adsorbent site will be stronger to metal ions with a smaller k value; hence, the adsorption process is faster and more efficient [35].

On the other hand, the kinetics adsorption for SB-M is more appropriate to follow pseudo 1st order kinetics model with R^2^ value of 0.9881 and qe of 150.385 mg/g. The value of maximum adsorption capacity was also similar to the experimental result (150.628 mg/g). The difference in adsorption kinetic models for SB-M and SB-MH can be related to the presence of an amine group on SB-MH. The amine groups on bicomposites affect the surface charge of biocomposites to become positive (–NH_3_^+^). The water solubility of amine is higher due to hydrogen bonds involved in these lone electron pairs, then interacting with Pb(II) ions in solution. The chemical adsorption occurred faster than that for SB-M. Therefore, the group of amines on biocomposites enhanced the capacity of adsorption toward Pb(II) ions.

### 2.3. Effect of pH on Pb(II) Ion Adsorption, Chemical Oxygen Demand (COD), Total Suspended Solid (TSS), and Dyes Adsorption 

The pH of solution can increase or inhibit the adsorption of ions by changing the properties of the adsorbent surface charge; pH solution is one of the significant factors to affect the adsorption. The Pb(II) ion adsorption process can be in the form of ion exchange, surface complex formation or electrostatic interactions between Pb(II) ions and the adsorbent [36]. Figure 5a shows the effect of pH on adsorption of Pb(II) ions. The adsorption process took place optimally for SB-M and SB-MH at pH of 5 with adsorption capacity of 148.379 and 208.302 mg/g, respectively. The concentration of hydrogen ions at solution with lower pH (~5) leads to a higher value and this causes hydrogen ions to contend with Pb(II) ions for the active site; consequently, the presence of hydrogen ions enhances the adsorption capacity of Pb(II) ions [37]. On the other hand, increasing pH value causes the gradual decrease in the hydrogen ion concentration and thus the adsorbent surface was deprotonated. At this condition, the bonding occurs between Pb(II) ions and OH- anions to form Pb(OH)_2_. The increase value of pH (>5.8) causes the Pb(II) ions to be hydrolyzed in solution and further configures hydroxides [37]. The highest capacity of Pb(II) ion adsorption occurs at pH 5 and subsequently decreases; the soluble Pb(OH)_2_ species become dominant and then precipitation of Pb(OH)_2_ occurs at pH above 6.5 [38]. The optimal adsorption process is carried out in acidic conditions; pH 5 is the optimal condition for the adsorption process in this study. This condition also could be applied for metal ion removal in waste water.

COD concentration is influenced by role of the pH solution; the adsorption phenomenon is correlated to the ion exchange process. pH adjustment of the solution influences the adsorbent surface charge, ionization degree, stability and color intensity of the compounds in solution [39]. Figure 5b shows COD reduction at various pH. The largest reduction of COD occurred at pH of 5 with a value of 71.29% and 73.63% for SB-M and SB-MH, respectively. Generally, COD value decreased with increasing pH. For higher pH, the lower adsorption occurs due to an increase in the diffusion barrier of organic ions as well as the abundance of OH- ions; this causes the competition between organic molecules and the adsorbent surface [40]. At low pH, the positively charged group of organic molecules is bound by the attraction of electrostatic forces to the negative charge of the adsorbent surface. This is in keeping with the previous studies where low pH solutions increased the COD reduction using an adsorbent made from bagasse [41].

Besides having an effect on COD reduction, the adsorbent also has capability to degrade TSS. Stable suspended particles can be neutralized through a destabilization process with the addition of a positive charge from the adsorbent, afterward the energy resistance is reduced and a flock core is formed. The existence of a positive charge that is able to absorb negative charges on suspended particles surface causes the repulsion force between suspended particles in the waste to weaken. The comparison of TSS reduction in differences of pH is presented in Figure 6a, and the decrease in TSS is not very significant for all adsorbents. Another effect on SB-M and SB-MH adsorption is dye reduction (Figure 6b). The high dye reduction for SB-M and SB-MH occurred at pH 6. Dye reduction number was obtained at about 92.56% and 96.69% for SB-M and SB-MH, respectively. The reduction color for dye was observed before adsorption and after adsorption by using SB-M and SB-MH (Figure 6b, inset). It is clear that the original color (purple) disappears, becoming a clear solution after adsorption. The decrease of dye intensity in solution is probably due to protonation that would be the specific cause for the enhancement of electrostatic interaction between the negatively charged anions in the molecule of dye and the positive charge of the adsorbent active site [42].

In fact, the hydrogen ion concentration in the solution with pH of 5 is higher and this causes the hydrogen ions to compete with another contaminant for the active site. Hence, the presence of hydrogen ions can reduce the adsorption capacity of dye. Although influenced by the pH in solution, in this study the decrease in dye intensity at various pH did not show significant difference. 

### 2.4. Reusability Performance of Amine Magnetic Sugarcane Bagasse Biocomposites 

In order to evaluate the regeneration of biocomposites, 0.01 M HCl was elected as the effluent for Pb(II) ion desorption. The regeneration using 0.01 M HCl provides better desorption for the reuse process. As reported, the optimum desorption efficiency of the Pb(II) ion was obtained at value of 91% with 0.01 HCl [43]. At low pH, the hydrogen ions were combined with the ligands, then took place the adsorption site of the Pb(II) ion. Spontaneously, the Pb(II) ion will be released into solution. Desorption using HCl allows Pb(II) ions to be able to bind Cl- to form PbCl_2_; thus, adsorbent can be reused for the next process of adsorption. The adsorption performance of Pb(II) ions using SB-M and SB-MH was conducted in four cycles (Figure 7). The desorption efficiency for SB-M and SB-MH was about 92.11% and 96.96%, respectively. For both adsorbents of SB-M and SB-MH, the capacity of the Pb(II) ion adsorption during the reusable process decreased to be only below 10% after 4th cycles, i.e., 7.89% and 3.84%, respectively. The decreasing of adsorption capacity of adsorbents in reusability process also occurred in previous studies for biocomposites [20]. Derivation occurs due to unalterable ions of partial functional groups on the surface of biocomposites, thus affecting the subtraction in the number of sites and weakening the electrostatic interaction between the adsorbate and adsorbent [44]. A slight subtraction in the capacity of adsorption is owing to the incomplete desorption of divalent metal ions from the surface of adsorbent and also due to the reduction of some surface functional groups such as -OH and the amine group as well as the leaching of weak inorganic materials (Fe_3_O_4_) [45].

This shows that the amine magnetic sugarcane bagasse biocomposites have potential as a reusable adsorbent. Based on this study, the SB-MH has a better ability than SB-M including the process of reusing biocomposites. In other words, SB-MH is a more effective adsorbent for wastewater treatment, including Pb(II) ion removal, reducing of COD, TSS and dye in aqueous solution. These advantages make amine magnetic biocomposites more effective and confirm their suitability as adsorbents for a wide range of pollutant in aqueous solutions. 

## 3. Materials and Methods

### 3.1. Materials

The sugarcane bagasse was procured from a sugarcane juice seller at Banjarbaru, South Kalimantan, Indonesia. Wastewater containing dye was obtained from Sasirangan textile industry in Banjarbaru, Indonesia. The sasirangan textile contains TSS and COD of about 0.252 and 38.46 mg/L, respectively. The color absorption intensity by UV was identified about 0.6 for 3 times dilution. Lead (II) nitrate ((Pb(NO_3_)_2_), iron trichloride hexahydrate (FeCl_3_.6H_2_O), ethylene glycol (C_2_H_6_O_2_), sodium acetate anhydrous (C_2_H_3_NaO_2_), hexamethylenediamine (C_6_H_16_N_2_), hydrochloric acid (HCl), ethanol (C_2_H_5_OH), and caustic soda (NaOH) were purchased from Sigma Aldrich (Singapore).

### 3.2. The Sugarcane Bagasse Delignification

Tap water was used for washing sugarcane bagasse (SB) and dried at 80 °C for 3 h in an oven. The dried SB was grinded to form SB powder which has 60 mesh in size. The process of delignification was performed by mixing 1% NaOH (*w*/*v*) and 45% *w*/*v* of SB in the flask at 80 °C for 2 h under stirring. After cooling at room temperature, SB were filtered and washed with deionized (DI) water until the filtrate becomes neutral pH. Finally, the delignified SB (SB-D) was obtained by drying for 6 h at 80 °C in an oven.

### 3.3. Synthesis of Amine Magnetic Sugarcane Bagasse Biocomposites

One of the methods for preparation of amine-functionalized magnetic nanoparticle biocomposites is solvothermal reaction [20]. Initially, 0.8 g of iron trichloride hexahydrate and 1.6 g of sodium acetate anhydrous were placed in a beaker glass and 0.5 g of SB-D dissolved in 24 mL of ethylene glycol was poured to the flask by intense stirring for 10 min at 50 °C. Further, 7 mL of hexamethylenediamine was added into mixture. Finally, the mixture solution was placed in a Teflon Stainless Steel Autoclave and underwent a reaction for 6 h at 200 °C. Further, the solution was cooled at a room temperature. An external magnet field was used to collect the biocomposites from the solution and then rinsed with DI water followed by ethanol three times. The obtained material (SB-MH) was stored in DI water for further use. As a control, magnetic sugarcane bagasse biocomposites without addition of hexamethylenediamine (SB-M) were also produced.

### 3.4. Adsorption of Pb(II) Ion, Total Suspended Solid (TSS), Chemical Oxygen Demand (COD) and Dye onto Amine Functionalized Magnetic Nanoparticles Biocomposites

The adsorption capacity of biocomposites was conducted in a batch adsorption experiment to adsorb Pb(II) ions using wastewater of textiles as a dilution with initial concentration of Pb(II) of 100 ppm. The contact time (15, 30, 60, 120, 180, 240 and 360 min) and pH (5, 6 and 7) effects on the adsorption process were observed comprehensively. The desired value of pH was obtained by adding of 0.1 M NaOH. A total of 200 mL of certain Pb(II) ion concentrations was added to a weight amount of adsorbent. Then, the solution was shaken for various times and pH at room temperature to get the equilibrium condition. The external magnetic field was used to separate the solution for 2 min. Atomic Adsorption Spectrophotometer (AAS) was detected to calculate the remaining concentration of Pb(II) ions. The duplicated data and average value were taken for this study. The capability of biocomposites for repeated use in adsorption was also investigated; desorption of the Pb(II) ion-loaded SB-MH and SB-M was done by shaking for 4 h in 0.1 N HCl. Regenerated SB-MH and SB-M were used as an adsorbent for the next cycle after washing with DI water until filtrate ~pH 7. The capacity of adsorption was determined by applying the equation:(3)$$q_{e}{=(C}_{o}-{\;C}_{e})\frac{V}{m}\;$$
where Co (mg L^−1^) is the concentration of beginning Pb(II) ion and Ce (mg L^−1^) is the equilibrium concentration of the Pb(II) ion. V (L) is the solution volume and m (g) is the amount of adsorbent. 

### 3.5. Characterization

A transmission electron microscopy (TEM) image was taken using a Hitachi H-800 transmission electron microscope (Tokyo, Japan). Surface morphology of the sample was tested by Field-emission scanning electron microscopy (FE-SEM, JEOL JSM-6500F, JEOL Ltd., Tokyo, Japan) where all samples were sputter-coated by platinum. The X-Ray Fluorescence (XRF) observation by the PANalytical/Minipal machine (Malvern, London, UK) was used for detecting elements in the sample. The X-ray diffraction (XRD) measurement was investigated using a Rigaku D/MAX-B X-ray diffractor (Rigaku Co., Tokyo, Japan) meter equipped with Copper K-alpha (CuKα) radiation. The operation voltage worked at 40 kV and the current for machine worked at 100 mA. Autosorb-1 instrument (Anton Paar Quanta Tec. Inc., Boynton Beach, FL, USA) was used to evaluate Brunauer–Emmet–Teller (BET) surface area equipped with the nitrogen adsorption–desorption using a Quantachrome. The magnetic property was investigated by the superconducting quantum interference device (SQUID, LakeShore 7307, Lake Shore Cryotronics Inc., Westerville, OH, USA) magnetometer. The functional groups on the sample were investigated by using Fourier transform infrared spectrometry (FT-IR, Bio-rad, Digilab FTS-3500Agilent Digilab, CA, USA). Thermal gravimetric analysis (TGA) was used to study the degradation of material (Perkin Elmer, Waltham, MA, USA) in N_2_ atmosphere flow 10 °C/min at 30–1000 °C. The degradation of the material component was studied via mass reduction from curves of TGA. 

### 3.6. Analysis 

The amine content on biocomposites was calculated by the retro-titration method [46]. The 50 mg of sample was put in 25 mL of 0.01 M HCl and shaken for 2 h at room temperature. After filtration, the filtrate was titrated by 0.01 N NaOH, and the calculation of amine group concentration was determined by applying the equation:(4)$$C_{NH_{2}}\left[\frac{\left(C_{\mathrm{HCl}}{\;\times \;V}_{\mathrm{HCl}}\right)-{\;(5\;C}_{\mathrm{NaOH}}{\;\times \;V}_{\mathrm{NaOH}})}{m_{\mathrm{sample}}}\right]\;$$
where C_HCl_ is the H_Cl_ solution concentration (mmol/L). C_NaOH_ is the NaOH solution concentration (mmol/L); volume of HCl solution (L) is notated as V_HCl_. V_NaOH_ is the volume of NaOH used in the titration of non-reacted acid’s excess (L); weight of sample (g) is notated as m_sample_.

COD analysis was conducted using the titrimetric permanganate method, as 100 mL of the sample was treated with 1 mL of 6N H_2_SO_4_ and followed by 10 mL of 0.01 N KMnO_4_. The mixture in the flask was heated at boiling point and cooled down to room temperature for 10 min, and further 10 mL of 0.01 N of H_2_C_2_O_4_.2H_2_O was added to the flask. The mixture was titrated by 0.01 KMnO_4_ until the color of solution become pink. The blank concentration using DI Water has to be measured as also titrated by 0.01 KMnO_4_. The titration continued until the purple color on the mixture disappeared. The COD concentration was calculated by the following equation:
(5)$$\mathrm{COD}\left(\frac{\mathrm{mg}}{L}\right)=\left[\left(\left(\left(a+b\right){\;\times \;C\;\mathrm{KMnO}}_{4}\right)-\left(\left(V\times C\right)H_{2}C_{2}O_{4}\right)\right)\right]\;\times \;8000$$
where a is titration standardization volume of KMnO_4_, b is titration sample volume of KMnO_4_, C KMnO_4_ is solution concentration (N), V H_2_C_2_O_4_ (L) and H_2_C_2_O_4_ are volume and concentration of H_2_C_2_O_4_ (N), respectively.

TSS was analyzed using the Standard Test Method for Filterable and Nonfilterable Matter in Water. The sample was well-mixed, and then the measured volume of a water sample is filtered through a pre-weighed glass fiber filter. The filter was heated at 104 ± 1 °C until constant mass and then weighed. The TSS was calculated by the following equation:
(6)$$\mathrm{TSS}\left(\frac{\mathrm{mg}}{L}\right)=\frac{\left(A\;-\;B\right)\;\times \;100}{C}$$
where A is dry weight of residue and filter (mg), B is dry weight of filter (mg), and C is sample volume (L). The dye intensity was tested by UV-Vis spectroscopy (V-550-JASCO, JASCO, Easto, MD, USA) which worked on wavelength maximum color adsorption on sample.

## 4. Conclusions

By incorporating amine magnetic with sugarcane bagasse to form a biocomposites, the material was successfully synthesized with high intensity of magnetic property. The utilization of sugarcane bagasse as biomass enhances the value added of waste material. The biocomposites showed excellent performances in Pb(II) ion adsorption, reduction of COD, TSS and dye in aqueous solution. In addition, the stable material tested for reusability was achieved with insignificant loss of performance. The synergic effects of amine magnetic sugarcane bagasse biocomposites on wastewater pollutant leads to this material having the potential to be developed and applied in waste water treatment. Therefore, the biocomposites are considered as one of the promising and attractive alternative materials for waste water cleaning and controlling environmental contamination.

## Data Availability

Not applicable.

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
