# Peer review of "The Amine Functionalized Sugarcane Bagasse Biocomposites as Magnetically Adsorbent for Contaminants Removal in Aqueous Solution"

_molecules, 2021, doi:10.3390/molecules26195867_

Round 1
Reviewer 1 Report
You can be a little bit more comprehensive in terms of representation if you want
Author Response
Dear Reviewer 1
Please see the attachment.

Reviewer 2 Report
The manuscript “The Amine Functionalized Sugarcane Bagasse Biocomposites as Magnetically Adsorbent for Contaminants Removal in Aqueous Solution”, Molecules manuscript ID 1348413 describes the synthesis of magnetic nanoparticle-functionalized bioderived materials for contaminant removal. The manuscript has enough experimental detail to be reproducible and describes the results adequately. There are a few questions or comments:
- Was the bagasse processed in any way for FE-SEM imaging? Was it sputter coated to provide a conductive layer?
- How were the magnetic nanoparticles extracted from the biocomposite for magnetic and TEM analysis? (Fig. 3a and 2c inset).
- When comparing performance of SB-M and SB-MH, the authors mention that the amine functionalization is responsible for the difference in performance. Did the authors measure the surface area of SB-M? Could differences in magnetic nanoparticle attachment or formation have an influence in the adsorption behavior?
- Does the dependence of absorption and COD on pH have any relationship to the pKa of the hydroxylamine?
- Can the authors explain if pH 5 is an acceptable pH level for use of this bioabsorbent in point of use or water treatment plants?
- How long was the magnetic separation step for Pb, COD, TSS and dye quantification?
- As per the discussion, acidic pH favors pollutant adsorption on the biocomposite. Then why is the Pb desorption experiment done in an acidic environment as well? If the dominant role for the desorption is the presence of Cl- ions, could NaCl or other source of Cl counterions be used?
- Would the same desorption method be used for dye degradation and reuse, or would a specific desorption method need to be developed for each contaminant?
- Was the effect of acidic pH evaluated for iron oxide degradation? SEM or magnetic characterization done after acidic pH incubation would be desirable, as the acidic pH is likely to dissolve partially the iron oxide phases.
- Was content of Fe in solution evaluated using AAS? Are there any interferences that should be considered for Pb or COD quantification that could come from Fe oxide dissolution?
- The absorption experiments were done in duplicate and the average taken of each experiment. Can the authors include the standard deviation of the two replicates in Figure 4 and 5a?
- What is the dye used for testing? Please indicate in manuscript, this is important information.
- COD measurement procedure not adequately described. Please include briefly the “Standard Test Method for Filterable and Nonfilterable Matter in Water” assay description
- The authors should put their work in the context of published work for magnetic functionalized bagasse for contaminant removal, for example Toledo-Jaldin et al. Environm. Sci. Poll. Res. 2020 Low-cost sugarcane bagasse and peanut shell magnetic-composites applied in the removal of carbofuran and iprodione pesticides; Pineres et al. Cellulose 2020 Magnetic paper from surgacane bagasse fibers modified with cobalt ferrite nanoparticles; Liang et al. Sci. Reports 2020 Removal of aqueous Cr(VI) by magnetic biochar derived from bagasse and others cited throughout the manuscript. Is the present work more efficient/more environmentally-friendly/more cost effective?
- Minor spelling and grammar revision needed for readability.
Author Response
Dear Reviewer 2
Please see the attachment

Reviewer 3 Report
The submitted ms is focused on lead and other contaminants adsorption by magnetically functionalized sugarcane bagasse. At first sight, the article looks relatively good, but after a closer examination, several serious shortcomings are evident.
- In the first sentence in Introduction authors mentioned the use of agriculture waste biomass as metal “absorbents”. I strongly disagree with such incorrect use of terminology in article: absorption vs adsorption; absorbent vs adsorbent (lines 30-31) , absorption capacity vs adsorption capacity (lines 220-222; 258) .
- Please explain the role of hydrophobic interactions and hydrogen bonding in metal binding by biomass (lines 32, 33).
- Overall, the introduction of the article is chaotic, not well organized and does not bring up-to-date findings.
- Does the prepared biocomposite really contain 99% Fe (line 92)? Compare with results from TGA.
- 4 – please add the initial Pb concentration.
- Although several precise analytical procedures were applied there is a lack of serious discussion supported by obtained data describing the mechanism of Pb adsorption by prepared biocomposite.
- The role of Fe304 in Pb removal is ignored. In aqueous solutions, −FeOH groups are present on iron oxide surfaces, and depending on the solution pH, they protonate or deprotonate to FeOH2+and FeO−. With the increase in the solution pH, FeO− increased, what can facilitate the removal of Pb2+ by electrostatic adsorption. I suppose that such synergistic interaction with the –FeOH groups of iron oxides caused better Pb adsorption. Please check the recently published article (https://www.mdpi.com/1996-1944/13/16/3619/htm ) where the role Fe3O4 in Co adsorption by magnetically modified biomass was confirmed by the elemental mapping.
- The title of section 2.3 should be rewritten.
- In line 258 authors stated that “the presence of hydrogen ions enhances the absorption (!!) capacity of Pb(II) ion”. Please explain.
- There is no information about the wastewater used. Therefore results regarding the reduction of TSS, COD and dye content are useless. I strongly advice authors to add basic characteristics of wastewater used.
- The reusability of biocomposites were also studied in four adsorption-desorption cycles. What was the desorption efficiency (%)? This should be at least mentioned in the text.
- Method for COD determination is missing in Materials and methods.
Author Response
Dear Reviewer 3
Please see the attachment

Round 2
Reviewer 3 Report
Authors tried to improve manuscript quality in revised version. However, I still feel that the article needs to be revised.
In my previous review I asked authors to explain the role of hydrophobic interactions and hydrogen bonding in metal binding by biomass. I hypothesized that by studying the relevant literature, the authors would find that the key mechanisms involved in metal sorption by biosorbents are ion exchange, complexation, chelation, and microprecipitation. Not hydrophobic interactions. I suggest revising the introduction using relevant sources.
Moreover, it is still true that the introduction of the article is chaotic, not well organized and does not bring up-to-date findings. The revisions implemented are insufficient.
I doubt that the prepared composite contains 99% Fe. TGA analysis indicates a weight decrease of more than 60% due to the decomposition of cellulose and hemicelluloses. This means that the Fe content can in no way represent 99% of the composite.
Please omit “hydrogen ions enhances the adsorption capacity of Pb ion.” If this is true, the highest sorption would occur at pH1...
Regarding the desorption studies. A crucial role in Pb desorption is played by competition with H+ ions, which displace Pb ions as a result of ion exchange. Overall, the discussion regarding desorption is confusing (lines 389-391) and does not provide relevant rationale.
Although the article presents interesting information, the presentation of the results is not at an acceptable level. I recommend that the authors critically evaluate the results and formulate and discuss the findings based on relevant data. I am not an english native speaker, but the English should be substantially improved.
